# Triple Antiplatelet Therapy with Cilostazol and Favorable Early Clinical Outcomes after Acute Myocardial Infarction Compared to Dual Antiplatelet Therapy with Standard or Potent P2Y12 Inhibitors

**DOI:** 10.3390/jcm11226856

**Published:** 2022-11-21

**Authors:** Sungwook Byun, Su Nam Lee, Sungmin Lim, Eun Ho Choo, Ik Jun Choi, Chan Joon Kim, Donggyu Moon, Mahn-Won Park, Chul Soo Park, Youngkeun Ahn, Myung-Ho Jeong, Kiyuk Chang

**Affiliations:** 1Division of Cardiology, Department of Internal Medicine, Bucheon St. Mary’s Hospital, College of Medicine, The Catholic University of Korea, Bucheon 14647, Republic of Korea; 2Division of Cardiology, Department of Internal Medicine, St. Vincent’s Hospital, College of Medicine, The Catholic University of Korea, Suwon 16247, Republic of Korea; 3Division of Cardiology, Department of Internal Medicine, Uijeongbu St. Mary’s Hospital, College of Medicine, The Catholic University of Korea, Uijeongbu 11765, Republic of Korea; 4Division of Cardiology, Department of Internal Medicine, Seoul St. Mary’s Hospital, College of Medicine, The Catholic University of Korea, Seoul 06591, Republic of Korea; 5Division of Cardiology, Department of Internal Medicine, Incheon St. Mary’s Hospital, College of Medicine, The Catholic University of Korea, Incheon 21431, Republic of Korea; 6Division of Cardiology, Department of Internal Medicine, Daejeon St. Mary’s Hospital, College of Medicine, The Catholic University of Korea, Daejeon 34943, Republic of Korea; 7Division of Cardiology, Department of Internal Medicine, Yeouido St. Mary’s Hospital, College of Medicine, The Catholic University of Korea, Seoul 11671, Republic of Korea; 8Division of Cardiology, Department of Internal Medicine, Chonnam National University Hospital, Gwangju 61469, Republic of Korea

**Keywords:** antiplatelet agent, myocardial infarction, mortality, bleeding

## Abstract

Current guidelines for the management of acute myocardial infarction (AMI) recommend potent P2Y12 inhibitors rather than clopidogrel to prevent ischemic events. However, their ischemic benefits are offset by an increased major bleeding risk. We compared the efficacy and safety of triple antiplatelet therapy with cilostazol in the first month after AMI. This study investigated 16,643 AMI patients who received percutaneous coronary intervention (PCI) with drug-eluting stents (DES) in nationwide, real-world, multicenter registries in Korea. Patients were divided into DAPT (aspirin and clopidogrel, *n* = 11,285), Triple (aspirin, clopidogrel and cilostazol, *n* = 2547), and Potent (aspirin and ticagrelor/prasugrel, *n* = 2811) groups. The primary outcomes were net adverse clinical events (NACE), a composite of death from any cause, myocardial infarction (MI), stroke, and TIMI major bleeding one month after AMI. After adjusting for covariates, there were no statistically significant differences in the risk of death from any cause, MI, or stroke between the three groups. However, the risk of TIMI major bleeding was significantly greater in the Potent group than in the DAPT and Triple groups (*p* < 0.001). Accordingly, NACE was significantly higher in the DAPT (HR 1.265; 95% CI 1.006–1.591, *p* = 0.044) and Potent groups (HR 1.515; 95% CI 1.142–2.011, *p* = 0.004) than in the Triple group. Triple antiplatelet therapy with cilostazol was associated with an improved net clinical outcome in the first month after AMI without increasing the risk of bleeding compared to potent or standard P2Y12 inhibitor-based DAPT.

## 1. Introduction

The current guidelines recommend dual antiplatelet therapy (DAPT) with one potent P2Y12 inhibitor and aspirin for at least 12 months to reduce ischemic event rates in patients with acute myocardial infarction (AMI) [1,2]. TRITON-TIMI 38 and PLATO trials showed that potent P2Y12 inhibitors, including prasugrel and ticagrelor, were superior to clopidogrel for the reduction of ischemic events in acute coronary syndrome patients [3,4]. Accordingly, patients with AMI undergoing percutaneous coronary intervention (PCI) are strongly recommended to take potent P2Y12 inhibitors preferentially over clopidogrel. However, a higher bleeding risk was observed with potent P2Y12 inhibitors compared to clopidogrel in these large, randomized trials, along with strong antiplatelet efficacy. Therefore, finding better antiplatelet strategies to achieve an optimal balance between ischemic and bleeding risks in AMI patients remains an important research goal.

The risk of ischemic events is highest in the early period following AMI but decreases over time [5,6]. The initial increased risk of ischemic events appears associated with pro-thrombotic factors, such as platelet activation, sympathetic activation, and inflammation [5,7]. However, the bleeding events observed with the use of potent P2Y12 inhibitors occur throughout their use [8,9,10]. Accordingly, a conventional DAPT with clopidogrel has some weakness in reducing early ischemic events after AMI, and a potent P2Y12 inhibitor-based DAPT strategy has some weakness in increasing bleeding events throughout the post-MI period although has strength in reducing ischemic events. Therefore, triple antiplatelet therapy composed of cilostazol, aspirin, and clopidogrel may have the potential to further reduce early ischemic events compared with clopidogrel and not increase the risk of bleeding compared with potent P2Y12 inhibitors in AMI patients. According to the results of analysis of 4203 ST-segment elevation MI (STEMI) patients who received primary PCI, triple antiplatelet therapy with cilostazol significantly reduced ischemic events without increasing major bleeding events compared with a conventional DAPT with clopidogrel [11]. To date, data on the efficacy and safety of triple antiplatelet therapy with cilostazol compared to DAPT with a potent or standard P2Y12 inhibitor in the earlier period of AMI does not exist. In a large, real-world AMI cohort undergoing PCI with the use of drug-eluting stents (DES), the best antiplatelet strategy to optimally adjust the balance between ischemic and bleeding complications was investigated.

## 2. Methods

### 2.1. Study Population

In the present study, the eligibility of AMI patients from the Korea AMI Registry-National Institutes of Health (KAMIR-NIH) and Cardiovascular Risk and Identification of Potential High-Risk Population in AMI (COREA-AMI) registries was assessed. The KAMIR-NIH registry is a prospective, multicenter, web-based observational cohort study for the prognostic evaluation of AMI patients at 15 centers in Korea [12]. The COREA-AMI registry was designed to analyze the real-world, long-term, clinical outcomes in all consecutive patients with AMI at nine major cardiac centers in Korea performing high-volume PCI. The COREA-AMI I registry included AMI patients who received PCI from January 2004 to December 2009, and the COREA-AMI II registry extended the follow-up period of COREA-AMI I patients and additionally enrolled AMI patients between January 2010 and August 2014. Epidemiologic, angiographic, and follow-up clinical outcome data of all AMI patients were consecutively registered in the electronic, web-based case report system. A total of 23,823 patients were assessed for eligibility (Figure 1). We excluded 3456 patients with duplicates in both registries, 2238 without medication information, 1047 without PCI procedure, and 439 with missing data. Patients were divided into three groups based on antiplatelet regimens: DAPT (aspirin and clopidogrel, *n* = 11,285, 67.8%), Triple (aspirin, clopidogrel, and cilostazol, *n* = 2547, 15.3%), and Potent (aspirin and ticagrelor or prasugrel, *n* = 2811, 16.9%). The KAMIR-NIH study and the COREA-AMI study were conducted in accordance with the Declaration of Helsinki. Independent statisticians at the Clinical Research Coordination Center managed the final dataset, and clinical research associates sealed it with code. All participants were provided written informed consent, and the study was approved by the Institutional Review Board at each participating center. The COREA-AMI registry is registered on ClinicalTrials.gov (study ID: NCT02806102).

### 2.2. PCI Procedure and Antiplatelet Medication

According to current standard guidelines, coronary angiography and PCI were performed. Antiplatelet regimen and periprocedural anticoagulation administration were performed in accordance with standard regimens. The loading dose of the antiplatelet agent (aspirin, 300 mg; clopidogrel, 300 mg or 600 mg; cilostazol, 200 mg; ticagrelor, 180 mg; or prasugrel, 60 mg) was prescribed for all patients before or during PCI. Patients with DES were prescribed P2Y12 inhibitor (clopidogrel, 75 mg once daily, or ticagrelor, 90 mg twice daily, or prasugrel, 10 mg once daily) with aspirin, 100 mg daily, for at least 12 months. At the discretion of the individual clinician, cilostazol (100 mg twice daily) was additionally prescribed as a triple antiplatelet therapy and recommended to be maintained for at least 1 month once started.

### 2.3. Follow-Up, Data Collection, and Analysis

Epidemiologic, angiographic, and follow-up clinical outcome data were collected using a web-based reporting system. If necessary, additional information was provided by viewing the medical records or by contacting by telephone. An independent clinical event adjudicating committee reviewed all data on outcomes reported from participating centers. Every patient in the KAMIR-NIH registry received clinical follow-ups of up to 3 years (median follow-up duration 36.4 months; interquartile range: 34.5–37.5 months) and every patient in the COREA-AMI II registry received clinical follow-ups as long as possible by 2019 (median follow-up duration 57.5 months; interquartile range: 33.7–86.5 months).

### 2.4. Clinical Outcomes and Definitions

Primary clinical outcomes were net adverse clinical events (NACEs) at 1 month after AMI, which included major adverse cardiac and cerebrovascular events (MACCEs) plus major bleeding events according to the Thrombolysis in Myocardial Infarction (TIMI) bleeding criteria. MACCEs included the composite of all-cause death, nonfatal MI, and stroke 1 month after AMI. AMI was defined by detecting an elevated cardiac biomarker value at least 1 higher than the 99th percentile of the upper limit, transient increases, and decreases, and at least one of the following clinical features: symptom of cardiac ischemia, new or new presumptive significant ST-segment T-wave change or new left bundle-branch block, pathologic Q wave development on electrocardiogram, imaging evidence of new abnormality in regional wall motion, or new loss of viable myocardium or intracoronary thrombus on angiography. ST-segment was defined as new ST-elevation at the J-point in two contiguous leads with the cut points [13]. Clinical presentation was divided into two groups: STEMI and NSTEMI. Stroke was defined as an episode of focal and global neurologic dysfunction related to brain, spinal cord, or retinal vascular injury as a result of infarction or hemorrhage.

### 2.5. Statistical Analysis

Continuous data were presented as the median and interquartile range or mean ± standard deviation and compared using analysis of variance, and multiple comparisons were assessed using the Bonferroni *t*-test. Categorical data were expressed as numbers and percentages and compared using the chi-square or Fisher’s exact test. Survival was performed using the Kaplan–Meier method and compared using the log-rank test. The impact of antiplatelet combination on survival was analyzed by Cox regression model. Multivariate Cox regression analyses used significant variables identified based on univariate Cox regression analyses (*p* < 0.05). The hazard ratio (HR) and 95% confidence interval (CI) were also calculated. A *p*-value < 0.05 was considered statistically significant. All statistical analyses were performed using Statistical Analysis Software (SAS, version 9.2, SAS Institute, Cary, NC, USA).

## 3. Results

### 3.1. Baseline Characteristics between Groups

Table 1 shows the baseline demographic, clinical, laboratory, and angiographic characteristics of patients classified based on the antiplatelet regimens: DAPT (aspirin and clopidogrel, *n* = 11,285), Triple (aspirin, clopidogrel, and cilostazol, *n* = 2547), and Potent (aspirin and ticagrelor or prasugrel, *n* = 2811). The prevalence of younger age, male, STEMI, and smoking history was significantly higher in the Potent group. The subjects had a higher body mass index, lower Killip classification, lower creatinine level, lower high-sensitivity C-reactive protein level, and higher low-density lipoprotein cholesterol level. In addition, the use of betablockers and statins was significantly higher in the Potent group. However, the prevalence of diabetes mellitus, hypertension, dyslipidemia, and previous MI history was significantly higher in the Triple group. The subjects had a higher glycated hemoglobin level and higher high-sensitivity C-reactive protein level. In addition, the use of renin-angiotensin system blockers was higher in the Triple group.

### 3.2. Clinical Outcomes

The incidence of MACCEs at the one-month follow-up was 4.6%, 3.1%, and 2.4% for the DAPT, Triple, and Potent groups, respectively. In addition, the incidence of TIMI major bleeding was 0.7%, 0.5%, and 1.9% for the DAPT, Triple, and Potent groups, respectively at the one-month follow-up. Accordingly, the incidence of NACEs at the one-month follow-up was 5.2%, 3.4%, and 4.2% for the DAPT, Triple, and Potent groups, respectively (Table 2). Regarding NACEs, MACCEs, and TIMI major bleeding, the differences between the three groups were significant within one month based on the Kaplan–Meier curve analysis (log rank *p* < 0.001, log rank *p* < 0.001, log rank *p* < 0.001, respectively, Figure 2). After adjusting covariates with a multivariate Cox hazard regression model, MACCEs at the one-month follow-up did not show statistically significant differences in the groups (Table 3). However, the risk of TIMI major bleeding was significantly higher in the Potent group than in the DAPT group (HR 3.043, 95% CI 2.119–4.369, *p* < 0.001) and the Triple group (HR 4.009, 95% CI 2.119–7.585, *p*< 0.001) at the one-month follow-up; risks between the DAPT and Triple groups were not significantly different. Accordingly, NACEs were significantly higher in the DAPT and Potent groups than in the Triple group at the one-month follow-up (DAPT; HR 1.265, 95% CI 1.006–1.591, *p* = 0.044; Potent; HR 1.515, 95% CI 1.142–2.011, *p* = 0.004) without a significant difference between the DAPT and Potent groups. After one month, differences were not observed in the MACCE, TIMI major bleeding, and NACE rates between Triple and Potent groups. However, compared with the Potent group, MACCEs and NACEs were significantly higher in the DAPT group. Propensity score matching analysis for the Triple and the Potent group was performed, and the results were the same as before (Table 4).

## 4. Discussion

The principal findings in the present study are as follows: Firstly, there were no statistically significant differences in MACCEs at the one-month follow-up regardless of antiplatelet regimens. Secondly, potent P2Y12 inhibitor-based DAPT significantly increased the bleeding events at one month compared with triple antiplatelet therapy and standard P2Y12 inhibitor-based DAPT without significant differences in bleeding between standard P2Y12 inhibitor-based DAPT and triple antiplatelet therapy. Finally, triple antiplatelet therapy with cilostazol was the most optimal antiplatelet strategy to create a balance between ischemic and bleeding events compared with potent or standard P2Y12 inhibitor-based DAPT in the earlier period after AMI.

During the early period after AMI, the ischemic events increase, but they decrease over time [5,6]. The initial increased ischemic risk seems to be associated with elevated prothrombotic factors [5,7]. Conversely, excessive major bleeding events observed with the use of potent P2Y12 inhibitors occur throughout their use [8,9,10]. In previous reports, AMI patients with major bleeding events confer sustained risks of both mortality and MACCEs [14,15]. Previous data have shown that even in clinical trial nonparticipants, bleeding events within 30 days were significantly higher than thereafter, and bleeding in the earlier period was independently associated with short-term mortality [16]. Therefore, in the early phase, AMI patients are vulnerable to both ischemic and bleeding events, and optimal antiplatelet strategies that reduce both ischemic and bleeding risks are needed.

Standard P2Y12 inhibitor-based DAPT has been widely used to prevent recurrent ischemic events and largely studied for acute coronary syndrome, including patients with AMI [17]. However, two randomized trials showed that potent P2Y12 inhibitors, such as ticagrelor or prasugrel, were more effective at preventing ischemic events than standard DAPT with clopidogrel in patients with acute coronary syndrome [3,4]. Based on these results, potent P2Y12 inhibitors have been advocated for preventing ischemic events in the guideline for acute coronary syndrome [2]. However, their ischemic benefits are offset by an increased risk of major bleeding. Furthermore, compared to Westerners, East Asians have unique features, such as a lower rate of ischemic events, a higher rate of bleeding events after PCI, and their response to antiplatelet medication [18,19,20]. In a retrospective analysis of Korean AMI patients, the potent P2Y12 inhibitors were associated with a significantly higher bleeding risk without ischemic benefits [20]. Recently, short-term DAPT strategies were proposed to reduce bleeding risk while maintaining ischemic benefits [21,22]. In the TWILIGHT and TICO trials, ticagrelor monotherapy after three months of DAPT was associated with a lower risk of major bleeding and cardiovascular events [21,22].

In several previous studies, triple antiplatelet therapy with cilostazol was suggested to have the potential to prevent both ischemic and bleeding complications in AMI patients. Triple antiplatelet therapy was associated with a reduction of in-stent restenosis, stent thrombosis, and MI at 12 months compared with standard DAPT [23,24]. Chen et al. demonstrated that triple antiplatelet therapy with cilostazol significantly lowered eight-month mortality without bleeding risks, even in patients with a high thrombotic risk of AMI [11]. Furthermore, in the ACCEL-AMI study, triple antiplatelet therapy with cilostazol in AMI patients undergoing PCI achieved greater inhibition of platelet aggregation compared with high-dose clopidogrel maintenance dose or standard DAPT, especially in subjects with CYP 2C19 loss-of-function allele [25]. Although cilostazol has proven benefits, triple antiplatelet therapy with cilostazol has fallen out of use with the advent of potent P2Y12 inhibitors.

Cilostazol has different mechanisms of action for various cells. First, cilostazol inhibits platelet aggregation and adhesion by inhibiting the expression of platelet activation markers [26,27]. In addition to antiplatelet function, cilostazol improves endothelial function via nitric oxide production and decreases various inflammatory responses within endothelial cells [28,29,30]. Furthermore, cilostazol promotes apoptosis for vascular smooth muscle cells and ultimately reduces neointimal hyperplasia [31]. Based on these pharmacological profiles, cilostazol was expected to have a positive role in coronary artery disease patients.

Recently, Kim et al. investigated the clinical outcomes between triple antiplatelet therapy with cilostazol and DAPT with potent P2Y12 inhibitors in Korean patients with AMI [32]. Similar to the results of our study, triple antiplatelet therapy with cilostazol resulted in fewer bleeding events than DAPT with potent P2Y12 inhibitors without significant differences in the clinical outcomes in their study. The author compared clinical outcomes at two years only between triple antiplatelet therapy with cilostazol and DAPT with potent P2Y12 inhibitor. However, in the present study, net clinical outcomes were investigated including MACCEs and TIMI major bleeding at one year of follow-up with a one-month landmark analysis among three different antiplatelet regimens in 16,643 AMI patients.

The present study had several limitations. First, because the present study consisted of nonrandomized, observational registries, unassessed confounding factors and selection bias could influence the outcomes. However, the present study had several advantages in minimizing the limitations of observational studies, such as multicenter design, large sample size, and statistical adjustments, including multivariable Cox regression. Second, there was no detailed information on drug adherence and persistence during follow-up in the KAMIR-NIH data. Finally, because the present study included only Korean AMI patients, expanding the results to other ethnic groups is not reasonable. Nonetheless, despite the limitations, the results emphasize the efficacy and safety of triple antiplatelet therapy with cilostazol in real-world practice. To the best of our knowledge, this is the first study in which the net adverse clinical outcomes of three different antiplatelet strategies in AMI patients were investigated.

## 5. Conclusions

Triple antiplatelet therapy with cilostazol was associated with favorable net clinical outcomes at the one-month follow-up after AMI in patients undergoing PCI with the use of DES without increasing bleeding risk compared with DAPT with potent or standard P2Y12 inhibitors. Therefore, triple antiplatelet therapy with cilostazol might be a safe and rational alternative to DAPT with potent P2Y12 inhibitors in AMI patients who have a high risk of bleeding, requiring complex and high-risk coronary intervention.

## Figures and Tables

**Figure 1 jcm-11-06856-f001:**
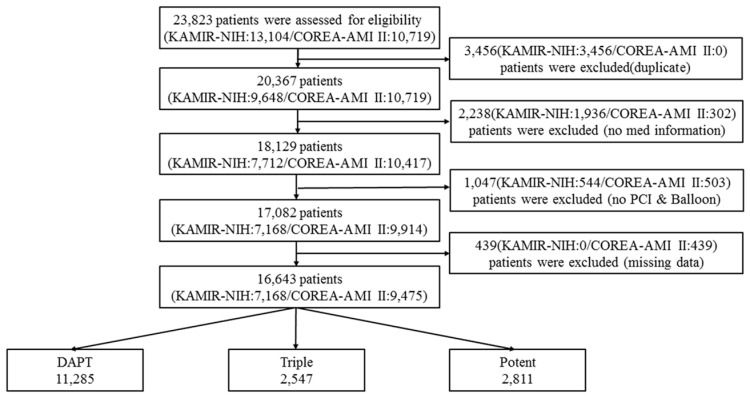
Flowchart of patients enrolled in the study.

**Figure 2 jcm-11-06856-f002:**
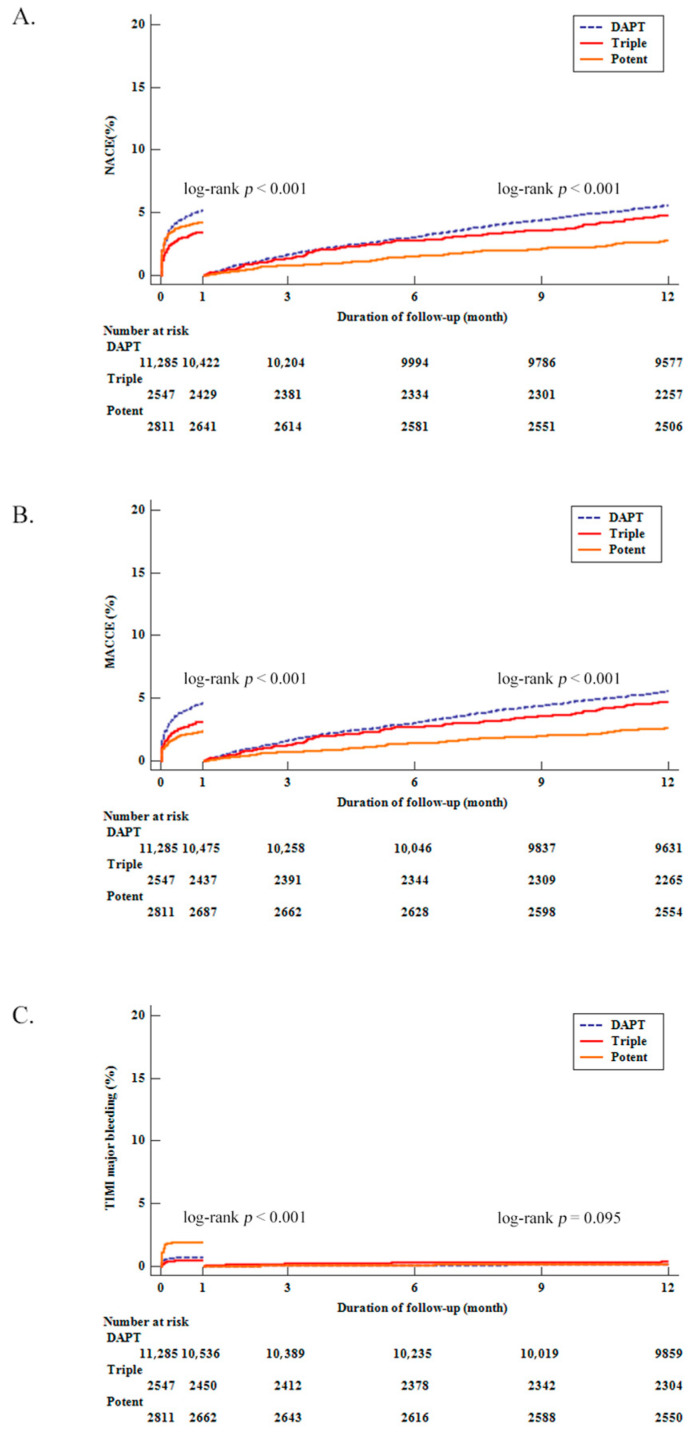
The twelve-month Kaplan–Meier curves and one-month landmark analysis of clinical outcomes based on antiplatelet combination. (**A**) NACEs (**B**) MACCEs (**C**) TIMI major bleeding.

**Table 1 jcm-11-06856-t001:** Baseline characteristics.

Variable	DAPT (*n* = 11,285)	Triple (*n* = 2547)	Potent (*n* = 2811)	*p*-Value	*Post-Hoc*
Male, *n* (%)	8153 (72.3)	1797 (70.6)	2307 (82.1)	<0.001	
Mean age (years)	64.0 ± 12.8	64.1 ± 12.6	60.1 ± 11.7	<0.001	1, 2 > 3
BMI	24.0 ± 3.2	24.1 ± 3.3	24.5 ± 3.2	<0.001	3 > 1, 2
Killip classification				<0.001	
1	8670 (76.8)	1983 (77.9)	2325 (82.7)		
2	977 (8.7)	216 (8.5)	155 (5.5)		
3	762 (6.8)	158 (6.2)	120 (4.3)		
4	876 (7.8)	190 (7.5)	211 (7.5)		
Final diagnosis				<0.001	
Non-STEMI	5184 (45.9)	1155 (45.4)	1103 (39.2)		
STEMI	6101 (54.1)	1392 (54.7)	1708 (60.8)		
Risk factors					
Family history of CAD	532 (4.7)	79 (3.1)	131 (4.7)	0.002	
Diabetes mellitus	3363 (29.8)	824 (32.4)	722 (25.7)	<0.001	
Hypertension	5798 (51.4)	1371 (53.8)	1282 (45.6)	<0.001	
Dyslipidemia	1533 (13.6)	434 (17.0)	434 (15.4)	<0.001	
Smoking, *n* (%)	4465 (39.6)	984 (38.6)	1356 (48.2)	<0.001	
Previous MI, *n* (%)	501 (4.4)	148 (5.8)	79 (2.8)	<0.001	
Laboratory finding					
HbA1C (%)	6.5 ± 1.5	6.8 ± 1.7	6.5 ± 1.5	<0.001	2 > 1, 3
Creatinine (mg/dL)	1.2 ± 1.2	1.1 ± 1.0	1.0 ± 0.7	<0.001	1, 2 > 3
hsCRP (mg/L)	2.3 ± 4.7	2.8 ± 5.3	1.4 ± 3.6	<0.001	2 > 1 > 3
Total cholesterol (mg/dL)	179.8 ± 44.5	181.4 ± 44.2	182.7 ± 43.8	0.008	3 > 1
Triglyceride (mg/dL)	129.1 ± 100.6	128.9 ± 94.6	137.7 ± 110.1	<0.001	3 > 1, 2
LDL cholesterol (mg/dL)	113.6 ± 38.4	115.4 ± 38.7	117.3 ± 38.7	<0.001	3 > 1
HDL cholesterol (mg/dL)	42.2 ± 11.3	41.9 ± 11.1	41.5 ± 10.8	0.013	1 > 3
Medication					
Aspirin	11,285 (100.0)	2547 (100.0)	2811 (100.0)		
Clopidogrel	11,285 (100.0)	2547 (100.0)	0 (0.0)		
Cilostazol	0 (0.0)	2547 (100.0)	0 (0.0)		
Ticagrelor	0 (0.0)	0 (0.0)	1754 (62.4)		
Prasugrel	0 (0.0)	0 (0.0)	1097 (39.0)		
Betablocker	9065 (80.3)	2118 (83.2)	2424 (86.2)	<0.001	
RAS blocker	8651 (76.7)	2090 (82.1)	2126 (75.6)	<0.001	
Statin	9938 (88.1)	2282 (89.6)	2667 (94.9)	<0.001	
LV EF (%)	52.1 ± 11.2	52.7 ± 11.4	52.9 ± 10.0	0.001	3 > 1
Target vessel				<0.001	
LAD	5478 (48.5)	1176 (46.2)	1298 (46.2)		
LCX	1851 (16.4)	415 (16.3)	468 (16.7)		
RCA	3686 (32.7)	838 (32.9)	980 (34.9)		
LM	261 (2.3)	118 (4.6)	65 (2.3)		
Others	9 (0.1)	0 (0.0)	0 (0.0)		
MVD	2711 (24.0)	930 (36.5)	625 (22.2)	<0.001	
Number of vessels	1.50 ± 0.71	1.53 ± 0.68	1.53 ± 0.72	0.134	
Total number of stents	1.2 ± 0.4	1.3 ± 0.5	1.1 ± 0.4	<0.001	2 > 1 > 3
Total stent length	28.3 ± 13.1	30.9 ± 15.8	28.7 ± 13.1	<0.001	2 > 1, 3
Mean stent diameter	3.1 ± 0.6	3.0 ± 0.6	3.1 ± 0.6	<0.001	1, 3 > 2

DAPT, dual antiplatelet therapy; BMI, body mass index; STEMI, ST-segment elevation myocardial infarction; CAD, coronary artery disease; MI, myocardial infarction; HbA1C, hemoglobin A1C; hsCRP, high-sensitivity C-reactive protein; LDL, low-density lipoprotein; HDL, high-density lipoprotein; RAS, renin-angiotensin system; LV EF, left ventricle ejection fraction; LAD, left anterior descending artery; LCX, left circumflex artery; RCA, right coronary artery; LM, left main; MVD, multivessel disease.

**Table 2 jcm-11-06856-t002:** Clinical outcomes.

Events	DAPT (*n* = 11,285)	Triple (*n* = 2547)	Potent (*n* = 2811)	*p*-Value
Clinical outcomes at one month				
MACCE, *n* (%)	513 (4.6)	78 (3.1)	66 (2.4)	<0.001
All cause death, *n* (%)	418 (3.7)	55 (2.2)	48 (1.7)	<0.001
Myocardial infarction, *n* (%)	47 (0.4)	19 (0.8)	10 (0.4)	0.057
Stroke, *n* (%)	70 (0.6)	17 (0.7)	12 (0.4)	0.429
TIMI major bleeding, *n* (%)	78 (0.7)	12 (0.5)	54 (1.9)	<0.001
NACE, *n* (%)	582 (5.2)	87 (3.4)	119 (4.2)	<0.001
Clinical outcomes between one and twelve months				
MACCE, *n* (%)	581 (5.2)	116 (4.6)	73 (2.6)	<0.001
All cause death, *n* (%)	406 (3.6)	86 (3.4)	50 (1.8)	<0.001
Myocardial infarction, *n* (%)	142 (1.3)	26 (1.0)	18 (0.6)	0.018
Stroke, *n* (%)	86 (0.8)	18 (0.7)	12 (0.4)	0.161
TIMI major bleeding, *n* (%)	17 (0.2)	9 (0.4)	4 (0.1)	0.081
NACE, *n* (%)	585 (5.2)	117 (4.6)	76 (2.7)	<0.001

DAPT, dual antiplatelet therapy; MACCE, major adverse cardio-cerebral event; TIMI, Thrombolysis in Myocardial Infarction; NACE, net adverse clinical event.

**Table 3 jcm-11-06856-t003:** Independent risk for clinical events by multivariate Cox hazard regression model.

Events	Adjusted Hazard Ratio	95% CI	*p*-Value
Clinical outcomes at one month			
MACCE			
DAPT vs. Triple (ref)	1.232	0.966–1.571	0.092
Potent vs. Triple (ref)	0.978	0.700–1.367	0.896
DAPT vs. Potent (ref)	1.259	0.969–1.639	0.085
TIMI major bleeding			
DAPT vs. Triple (ref)	1.318	0.714–2.433	0.378
Potent vs. Triple (ref)	4.009	2.119–7.585	<0.001
Potent vs. DAPT (ref)	3.043	2.119–4.369	<0.001
NACE			
DAPT vs. Triple (ref)	1.265	1.006–1.591	0.044
Potent vs. Triple (ref)	1.515	1.142–2.011	0.004
DAPT vs. Potent (ref)	0.835	0.681–1.024	0.083
Clinical outcomes between one and twelve months			
MACCE			
DAPT vs. Triple (ref)	1.186	0.969–1.453	0.099
Potent vs. Triple (ref)	0.815	0.605–1.098	0.179
DAPT vs. Potent (ref)	1.456	1.136–1.862	0.003
TIMI major bleeding			
DAPT vs. Triple (ref)	0.466	0.203–1.069	0.071
Potent vs. Triple (ref)	0.597	0.178–1.999	0.403
Potent vs. DAPT (ref)	1.281	0.419–3.916	0.664
NACE			
DAPT vs. Triple (ref)	1.190	0.973–1.456	0.091
Potent vs. Triple (ref)	0.860	0.641–1.154	0.314
DAPT vs. Potent (ref)	1.383	1.086–1.764	0.009

Adjusted by age, sex, BMI, Killip classification, final diagnosis, family history of CAD, diabetes mellitus, hypertension, dyslipidemia, smoking history, previous MI, HbA1C, hsCRP, creatinine, total cholesterol, triglyceride, LDL cholesterol, HDL cholesterol, betablocker, RAS blocker, Statin, LV EF, Target vessel, MVD, total number of stent, total stent length, mean stent diameter, MACCE, (major adverse cardio-cerebral event), DAPT (dual antiplatelet therapy), TIMI (Thrombolysis in Myocardial Infarction), and NACE (net adverse clinical event).

**Table 4 jcm-11-06856-t004:** Clinical outcomes and independent risk for clinical events by propensity score matching model.

Events	Triple (*n* = 1827)	Potent (*n* = 1827)	*p*-Value	HR (Potent vs. Triple (Ref))	95% CI	*p*-Value
Clinical outcomes at one month						
MACCE, *n* (%)	40 (2.2)	58 (3.2)	0.079	1.467	0.980–2.194	0.062
All cause death, *n* (%)	28 (1.5)	43 (2.4)	0.086	1.555	0.966–2.503	0.069
Myocardial infarction, *n* (%)	9 (0.5)	9 (0.5)	>0.999	1.013	0.402–2.553	0.978
Stroke, *n* (%)	7 (0.4)	9 (0.5)	0.804	1.297	0.483–3.482	0.606
TIMI major bleeding, *n* (%)	8 (0.4)	33 (1.8)	<0.001	4.163	1.923–9.012	<0.001
NACE, *n* (%)	47 (2.6)	90 (4.9)	<0.001	1.946	1.367–2.769	<0.001
Clinical outcomes between one and twelve months						
MACCE, *n* (%)	64 (3.5)	66 (3.6)	0.929	1.051	0.745–1.482	0.777
All cause death, *n* (%)	44 (2.4)	46 (2.5)	0.916	1.065	0.704–1.609	0.767
Myocardial infarction, *n* (%)	16 (0.9)	16 (0.9)	>0.999	1.017	0.509–2.035	0.961
Stroke, *n* (%)	13 (0.7)	10 (0.6)	0.664	0.784	0.344–1.788	0.563
TIMI major bleeding, *n* (%)	4 (0.2)	4 (0.2)	>0.999	1.029	0.257–4.116	0.967
NACE, *n* (%)	64 (3.5)	69 (3.8)	0.725	1.112	0.792–1.563	0.539

Adjusted by age, sex, BMI, Killip classification, final diagnosis, family history of CAD, diabetes mellitus, hypertension, smoking history, previous MI, HbA1C, hsCRP, creatinine, triglyceride, LDL cholesterol, betablocker, RAS blocker, Statin, LV EF, Target vessel, number of vessels, total number of stents, total stent length, mean stent diameter.

## Data Availability

The datasets used and/or analyzed during the current study are available from the corresponding author on reasonable request.

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
