# Peer review of "Triple Antiplatelet Therapy with Cilostazol and Favorable Early Clinical Outcomes after Acute Myocardial Infarction Compared to Dual Antiplatelet Therapy with Standard or Potent P2Y12 Inhibitors"

_jcm, 2022, doi:10.3390/jcm11226856_

Round 1
Reviewer 1 Report
This manuscript compared the efficacy and safety of triple antiplatelets therapy including cilostazol to other regimen in ACS and suggested the superiority of triple antiplatelets therapy to other antiplatelet regimens. This suggestion is interest but several concerns.
Clinical outcomes, NACEs, MACCEs, and bleeding events were assessed at 3 months after onset. However, Cilostazol was prescribed for at least 1 month. Authors should show whether these clinical outcomes occurred during prescribing Cilostazol.
Authors stated “Cilostazol (100 mg twice daily) was additionally prescribed as a triple antiplatelet therapy at the discretion of the individual clinician and recommended for at least 1 month once started”. How many patients met this recommendation of 1 month prescribing? And, prescribing duration of Cilostazol should be shown.
Lack of definition regarding STEMI in Methods section.
Author Response
Dear Editor-In-Chief and Reviewer(s) of Journal of Clinical medicine:
Thank you very much for your prompt review and for the helpful comments about our manuscript. We also appreciate the opportunity to resubmit our revised manuscript entitled “Triple antiplatelet therapy with cilostazol and favorable early clinical outcomes after acute myocardial infarction compared to dual antiplatelet therapy with standard or potent P2Y12 inhibitors”. In accordance with the respectful comments suggested by reviewer(s), we have revised our manuscript. By responding fully to their concerns, the manuscript has been significantly strengthened.
Reviewer 1
This manuscript compared the efficacy and safety of triple antiplatelets therapy including cilostazol to other regimen in ACS and suggested the superiority of triple antiplatelets therapy to other antiplatelet regimens. This suggestion is interest but several concerns.
1. Clinical outcomes, NACEs, MACCEs, and bleeding events were assessed at 3 months after onset. However, Cilostazol was prescribed for at least 1 month. Authors should show whether these clinical outcomes occurred during prescribing Cilostazol.
-> Thank you for the review point. As like the reviewer’s comments, we analyzed the clinical outcomes at 1 month and 1 year, and revised overall manuscripts (featured by “tracking changes”).
Authors stated “Cilostazol (100 mg twice daily) was additionally prescribed as a triple antiplatelet therapy at the discretion of the individual clinician and recommended for at least 1 month once started”. How many patients met this recommendation of 1 month prescribing? And, prescribing duration of Cilostazol should be shown.
-> Thank you for the review point. We excluded 2,238 patients without medication information. In addition, only those who used Cilostazol for at least 1 month were included in the triple group. However, duration of Cilostazol is not known because the medication information is checked at 1 month, 1 year, 2 year and 3 year (featured by “red color”).
- Lack of definition regarding STEMI in Methods section.
-> Thank you for the review point. As like the reviewer’s comments, we revised the part of “Clinical outcomes and definitions.” (featured by “tracking changes” on page 8-9).
I hope that this manuscript will be reviewed positively again from “Journal of Clinical medicine”.
Thank you very much.
Corresponding author : Su Nam Lee, MD. PhD
E-mail : [email protected]
Department of Internal Medicine, St. Vincent’s Hospital, The Catholic University of Korea, South Korea

Reviewer 2 Report
I have reviewed the manuscript ” Triple antiplatelet therapy with cilostazol and favorable early clinical outcomes after acute myocardial infarction compared with dual antiplatelet therapy with clopidogrel or potent P2Y12 inhibitors” by Byun et al. In this observational study, the authors use data from two Korean registries in order to identify whether triple antiplatelet treatment (Clopidogrel+ASA+ Cilostazol) is superior to DAPT (using ASA+Clopidogrel) or Potent (DAPT with more potent P2Y12 inhibitors and ASA) to prevent net adverse clinical events (MACCE+TIMI bleeding) after PCI and DES at 3 months. There is no difference in MACCE between “triple” and “potent” treatment groups, but there are more bleedings in the “potent” group, which makes the authors to suggest that “triple antiplatelet therapy with cilostazol might be a safe option to DAPT with potent P2Y12 inhibitors in AMI patients who have a high risk of bleeding requiring complex and high-risk coronary intervention”. Unfortunately, there are several issues with the study design that need to be addressed:
1) Cilostazol was selectively prescribed as an addition to DAPT, based on the clinician´s discretion. That is problematic due to selection bias. Which criteria was used for either therapy?
2) How is the data in the two registries validated? Any reference supporting the validation process? Even if the registries are validated, are the parameters used in this study specifically validated?
3) The treatment groups are not matched. There are significantly more STEMI patients in the “Potent”group than in the other two. This is problematic for several reasons.
a) STEMI patients are acute patients, in need of urgent PCI. There are
always more complications, especially bleeding problems, when a
procedure is performed under such stressful situation. The increased
TIMI bleeding could be explained by this fact.
b) The Triple and DAPT cohorts are mainly NSTEMI patients with
higher prevalence of normal cardiovascular risk factors like diabetes,
hypertension, previous MI etc. Usually NSTEMI patients have more
severe coronary artery disease than STEMI patients, which
complicates direct comparison between the groups.
Based on the identified problems and the fact that there is no difference in MACCE between Triple and Potent groups, except for TIMI bleeding, I suggest that you perform propensity score matched analysis. This strategy makes it possible to compare STEMI and NSTEMI patients separately and also match for other covariates. With this strategy, you will get an understanding if it is a real difference in favor for the Cilostazol triple antiplatelet treatment regimen.
Author Response
Dear Editor-In-Chief and Reviewer(s) of Journal of Clinical medicine:
Thank you very much for your prompt review and for the helpful comments about our manuscript. We also appreciate the opportunity to resubmit our revised manuscript entitled “Triple antiplatelet therapy with cilostazol and favorable early clinical outcomes after acute myocardial infarction compared to dual antiplatelet therapy with standard or potent P2Y12 inhibitors”. In accordance with the respectful comments suggested by reviewer(s), we have revised our manuscript. By responding fully to their concerns, the manuscript has been significantly strengthened.
Reviewer 2.
I have reviewed the manuscript “Triple antiplatelet therapy with cilostazol and favorable early clinical outcomes after acute myocardial infarction compared with dual antiplatelet therapy with clopidogrel or potent P2Y12 inhibitors” by Byun et al. In this observational study, the authors use data from two Korean registries in order to identify whether triple antiplatelet treatment (Clopidogrel+ASA+ Cilostazol) is superior to DAPT (using ASA+Clopidogrel) or Potent (DAPT with more potent P2Y12 inhibitors and ASA) to prevent net adverse clinical events (MACCE+TIMI bleeding) after PCI and DES at 3 months. There is no difference in MACCE between “triple” and “potent” treatment groups, but there are more bleedings in the “potent” group, which makes the authors to suggest that “triple antiplatelet therapy with cilostazol might be a safe option to DAPT with potent P2Y12 inhibitors in AMI patients who have a high risk of bleeding requiring complex and high-risk coronary intervention”. Unfortunately, there are several issues with the study design that need to be addressed:
Cilostazol was selectively prescribed as an addition to DAPT, based on the clinician´s discretion. That is problematic due to selection bias. Which criteria was used for either therapy?
-> Thank you for the review point. Cilostazol was additionally prescribed as triple antiplatelet therapy at the discretion of the individual clinician. Both KAMIR-NIH and COREA-AMI registry are observational study. In our study, the prevalence of diabetes mellitus, hypertension, dyslipidemia, previous MI history and MVD was significantly higher in the Triple group. And the Triple group had a higher level of glycated hemoglobin and high-sensitivity C-reactive protein. The subjects had a longer stent length and a greater number of stents. Although the Triple group had more poor prognostic factors, there were no statistically significant differences in the MACCEs at the 1 month follow-up regardless of antiplatelet regimens. In addition, we performed propensity score matching analysis for the Triple and the Potent group, and the results were the same as before. The table is shown at the bottom. (Table 4, featured by “tracking changes” on page 24). And we described “selection bias” in the limitation part. (featured by “tracking changes” on page 14).
How is the data in the two registries validated? Any reference supporting the validation process? Even if the registries are validated, are the parameters used in this study specifically validated?
-> Thank you for the review point. As like the reviewer’s comments, we added some sentences in method part. “Independent statisticians at the Clinical Research Coordination Center managed the final dataset and clinical research associates sealed with a code.” (featured by “tracking changes” on page 7).
- The treatment groups are not matched. There are significantly more STEMI patients in the “Potent”group than in the other two. This is problematic for several reasons.
- a) STEMI patients are acute patients, in need of urgent PCI. There are always more complications, especially bleeding problems, when a procedure is performed under such stressful situation. The increased TIMI bleeding could be explained by this fact.
b) The Triple and DAPT cohorts are mainly NSTEMI patients with higher prevalence of normal cardiovascular risk factors like diabetes, hypertension, previous MI etc. Usually NSTEMI patients have more severe coronary artery disease than STEMI patients, which complicates direct comparison between the groups.
->Thank you for the review point. To solve this problem, propensity score matching analysis was performed. However, the results were the same as before. The table is shown at the bottom (Table 4, featured by “tracking changes” on page 24).
- Based on the identified problems and the fact that there is no difference in MACCE between Triple and Potent groups, except for TIMI bleeding, I suggest that you perform propensity score matched analysis. This strategy makes it possible to compare STEMI and NSTEMI patients separately and also match for other covariates. With this strategy, you will get an understanding if it is a real difference in favor for the Cilostazol triple antiplatelet treatment regimen.
-> Thank you for the review point. As like the reviewer’s comments, the propensity score matched analysis was performed, however the results were the same as before. The table is presented below (Table 4, featured by “tracking changes” on page 24).
Table 4. Clinical outcomes and independent risk for clinical events by propensity score matching model
Events |
Triple (n=1,827) |
Potent (n=1,827) |
p-Value |
HR (potent vs triple (ref)) |
95% CI |
p-Value |
Clinical outcomes at 1 months |
||||||
MACCE, n (%) |
40(2.2) |
58(3.2) |
0.079 |
1.467 |
0.980-2.194 |
0.062 |
All cause death, n (%) |
28(1.5) |
43(2.4) |
0.086 |
1.555 |
0.966-2.503 |
0.069 |
Myocardial infarction, n (%) |
9(0.5) |
9(0.5) |
>0.999 |
1.013 |
0.402-2.553 |
0.978 |
Stroke, n (%) |
7(0.4) |
9(0.5) |
0.804 |
1.297 |
0.483-3.482 |
0.606 |
TIMI major bleeding, n (%) |
8(0.4) |
33(1.8) |
<0.001 |
4.163 |
1.923-9.012 |
<0.001 |
NACE, n (%) |
47(2.6) |
90(4.9) |
<0.001 |
1.946 |
1.367-2.769 |
<0.001 |
Clinical outcomes between 1 and 12 months |
||||||
MACCE, n (%) |
64(3.5) |
66(3.6) |
0.929 |
1.051 |
0.745-1.482 |
0.777 |
All cause death, n (%) |
44(2.4) |
46(2.5) |
0.916 |
1.065 |
0.704-1.609 |
0.767 |
Myocardial infarction, n (%) |
16(0.9) |
16(0.9) |
>0.999 |
1.017 |
0.509-2.035 |
0.961 |
Stroke, n (%) |
13(0.7) |
10(0.6) |
0.664 |
0.784 |
0.344-1.788 |
0.563 |
TIMI major bleeding, n (%) |
4(0.2) |
4(0.2) |
>0.999 |
1.029 |
0.257-4.116 |
0.967 |
NACE, n (%) |
64(3.5) |
69(3.8) |
0.725 |
1.112 |
0.792-1.563 |
0.539 |
*Adjusted by age, sex, BMI, killip classification, final diagnosis, family history of CAD, diabetes mellitus, hypertension, smoking history, previous MI, HbA1C, hsCRP, creatinine, triglyceride, LDL cholesterol, beta blocker, RAS blocker, Statin, LV EF, Target vessel, number of vessel, total number of stent, total stent length, mean stent diameter
I hope that this manuscript will be reviewed positively again from “Journal of Clinical medicine”.
Thank you very much.
Corresponding author : Su Nam Lee, MD. PhD
E-mail : [email protected]
Department of Internal Medicine, St. Vincent’s Hospital, The Catholic University of Korea, South Korea

Round 2
Reviewer 2 Report
Dear authors
The revised version of the manuscript has improved several levels and the propensity score results support your initial finding. I have no more comments the content of the study design or conclusions.